



# Cavity-enhanced photoacoustic sensor based on a whispering-gallery-mode diode laser

Yufeng Pan,[1,2] Lei Dong,[1,2*,] Hongpeng Wu,[1,2] Weiguang Ma[1,2] Lei Zhang[1,2], Wangbao Yin[1,2], Liantuan Xiao[1,2], Suotang Jia[1,2] and Frank K. Tittel[3]

[1] State Key Laboratory of Quantum Optics and Quantum Optics Devices, Institute of Laser Spectroscopy, Shanxi University, Taiyuan 030006, China

[2] Collaborative Innovation Center of Extreme Optics, Shanxi University, Taiyuan 030006, China

[3] Department of Electrical and Computer Engineering, Rice University, Houston, Texas 77005, USA

*Correspondence to*: Lei Dong (donglei@sxu.edu.cn)

**Abstract.** A cavity-enhanced photoacoustic (CEPA) sensor was developed based on an ultra-narrow linewidth whispering-gallery-mode (WGM) diode laser. A cavity-enhanced photoacoustic module (CEPAM) was designed to match the output beam from WGM-diode laser, resulting in an increase of the excitation light power, which, in turn, significantly enhanced the photoacoustic signal amplitude. The results show that a signal gain factor of 166 was achieved, which is in excellent agreement with the power enhancement factor of 175 after considering the power transmissivity. The performance of the sensor was evaluated in terms of the detection sensitivity and linearity. A $1\sigma$ detection limit of 0.45 ppmV for $C_2H_2$ detection was obtained at atmospheric pressure with a 1-s averaging time.

## 1 Introduction

Photoacoustic spectroscopy (PAS) is an important trace gas detection technique that is widely applied to atmospheric science, breath analysis and industrial process control (Yin et al., 2017; Wojtas et al., 2014; Yin et al., 2017). In the PAS technique, modulated excitation light is selectively absorbed by a target gas and results in the generation of an acoustic wave by non-radiative energy relaxation processes. One of the unique PAS advantages is that its sensitivity is proportional to excitation laser power and thus the performance of PAS sensors can be improved by increasing the excitation laser power. When a commercially available telecommunication diode laser is employed as an excitation light source, a commonly used method to increase the laser power is to use an erbium-doped fiber amplifier (EDFA) to boost the output optical power (Peng et al., 2009; Chen et al., 2018; He et al., 2018). For example, in 2015, H. Wu *et al.* demonstrated a quartz-enhanced photoacoustic spectroscopy (QEPAS) based $H_2S$ sensor operating at 1,582nm combined with an EDFA (Wu et al., 2015). With a ~1.4 W optical excitation power and 67 s averaging time, the $H_2S$ detection sensitivity was reduced from several ppmV level down to 142 ppbV in $N_2$, which is the best value for the $H_2S$ QEPAS sensors reported so far. X. Yin *et al.*, in 2017, developed a conventional photoacoustic (CPA) sensor operating at the same wavelength for $H_2S$ detection in $SF_6$ by means of an EDFA and a background-gas-induced high-$Q$ photoacoustic cell (Yin et al., 2017). A $1\sigma$ detection limit of 109 ppbV was achieved



with a 1-s averaging time. An alternative method of achieving high power is to combine PAS with cavity-enhanced absorption spectroscopy (CEAS) (Hippler et al., 2010; Kachanov et al., 2013; Wojtas et al., 2017). In 2014, P. Patimisco *et al.* proposed an intra-cavity QEPAS (I-QEPAS) sensor for $CO_2$ detection at 4.33 μm (Patimisco et al., 2015). A power enhancement factor of ~240 was achieved with an intracavity power of ~0.72 W, resulting in a minimum detection limit of

300 pptV at a total gas pressure of 50 mbar with a 20-s integration time.

Cavity-enhanced absorption spectroscopy (CEAS) is based on the use of optical resonant cavities in order to enhance light interaction with a gas species inside the cavity (Gherman et al., 2002; He et al., 2017; He et al., 2018). In CEAS setups, a proper locking between the laser wavelength and the cavity resonance mode must be carried out via two approaches: (i) the cavity length is controlled by a piezo transducer (PZT) for the resonance mode to follow laser wavelength; and (ii) the length

of the cavity is fixed, and the laser wavelength is locked to the cavity resonance mode. When the locking between the laser wavelength and the fundamental optical mode of the cavity is realized, the power inside the cavity will be enhanced significantly by a power enhancement factor $G$, and the value of $G$ can be calculated according to Eq.(1)

$$G = \frac{F}{\pi} \qquad\qquad (1)$$

where $F$ represents the finesse of the optical cavity.

In this manuscript, we developed a cavity-enhanced photoacoustic (CEPA) sensor system for acetylene ($C_2H_2$) detection at 1530.98 nm based on an ultra-narrow linewidth WGM-diode laser. A Fabry–Perot (F-P) cavity with a finesse of 550 was designed. The laser wavelength was locked to the F-P cavity mode by a Pound–Drever–Hall (PDH) locking technique (Drewer et al., 1983; Black., 2000) since the WGM-diode laser linewidth is far narrower than that of cavity mode. A differential photoacoustic cell with two electret condenser cylindrical microphones was designed to detect the photoacoustic

signal. To enhance the photoacoustic signal, the photoacoustic cell was inserted between the two cavity mirrors. The performance of the CEPA sensor was compared with a CPA sensor without cavity and evaluated at different $C_2H_2$ concentration levels.

## 2 Sensor design

### 2.1 Characterization of ultra-narrow linewidth WGM-diode laser

The portable WGM-diode laser (OEwaves, USA, OE4023-153160-PA02-FM100) has an ultra-narrow linewidth of <200 Hz for 10 μs and good tunability in the wavelength range of 1529 to 1532 nm with an output power of 4 mW after stabilizing. The laser wavelength can be tuned mode hop-free over 50 GHz at a tuning rate of 1 GHz/s by changing the temperature of the laser internal resonator and over 1 GHz at a tuning rate of 40 MHz/μs by an external voltage to change the length of an internal piezoelectric crystal. Figure 1 shows the tunable wavelength range of the WGM-diode laser for seven different

wavelength bands when the laser temperature was fixed and the internal piezoelectric crystal was changed by an external voltage. Due to the ~ 1-GHz limit for each wavelength band, the tunable range cannot link up end-to-end. The laser beam was collimated by a fiber-coupled collimator (OZ Optics, Canada, HPUCO-T.3A-1550-P-2AS). The output beam quality





from the collimator was evaluated by a scanning-slit optical beam profiler (THORLABS, USA, BP209-IR2/M). Figure 2(a) and (b) show the two-dimensional intensity distribution of the laser spot and the three-dimensional laser beam profile, respectively, at 15.2 cm from the collimator. The laser beam exhibits an excellent Gaussian fundamental mode with a spot size of 860 μm.

## 2.2 Design of the cavity-enhanced photoacoustic module

A cavity-enhanced photoacoustic module (CEPAM) consists of a differential photoacoustic cell, a F-P cavity, and a gas chamber with a gas inlet and outlet. The differential photoacoustic cell was designed as shown in Fig. 3(a), which resembles the well-known differential Helmholtz resonator (Zeninari et al., 1999; Starecki et al., 2014; Zheng et al., 2017). It has two identical 90 mm parallel tube-shaped channels with diameters of 8 mm as two acoustic resonators. Two buffer volumes with lengths of 10 mm and diameters of 20 mm connect to the two channels at both ends, thus making the two channels act as acoustic open-open resonators and create a total optical absorption length of 110 mm. This allows the beam from the WGM-diode laser to pass through the differential photoacoustic cell easily. When the laser intensity is modulated at the resonance frequency of the photoacoustic cell, a standing sound wave generated with the absorption of a target gas has its maximum acoustic pressure in the middle of the acoustic resonator. Hence, two selected electret condenser cylindrical microphones which have the same frequency response sensitivities are installed on the walls in the middle of each resonator to detect the acoustic pressure. The gas flow noise and external acoustic disturbances can be effectively suppressed by using a custom transimpedance differential preamplifier to amplify the signal from the two microphones, due to the fact that only one of the two acoustic resonators is excited by the laser, Therefore, the performance of the PAS cell is improved. Figure 4 shows the frequency response curves of the differential photoacoustic cell. The resonance frequency of the differential photoacoustic cell in air was $f_0 = 1781.0$ Hz and the full width at half maximum (FWHM) of the frequency response curves (resonance width) was $\Delta f = 40$ Hz, corresponding to a quality factor $Q = f_0/\Delta f = 45$.

Figure 3(b) illustrates the schematic of the CEPAM. The F-P cavity consists of a 25.4-mm plane mirror as the incident mirror and a 25.4-mm plane-concave mirror with a 1-m radius of curvature as the exit mirror. The two mirrors were coated with a high reflective coating of 0.995. The cavity length was 160 mm, which was longer than the 110-mm length of the differential photoacoustic cell. The designed differential photoacoustic cell was inserted between the two cavity mirrors as shown in Fig. 3(b). A gas chamber made of polymethyl methacrylate with a gas inlet and outlet was fabricated and used to provide environmental stabilization of the cavity.

## 3 Experimental setup of sensor system

The experimental setup of the CEPA sensor system is depicted in Fig. 5. A mode-matching len (L1), a half wave plate (λ/2), a polarization beam splitter (PBS) and a quarter wave plate (λ/4) were placed in front of the cavity. Two focusing lenses (L2 and L3) were used to focus the transmitted and reflected lights from the cavity onto the photodiode detectors (PD1 and PD2)





(THORLABS, USA, PDA10CF-EC), respectively. In order to achieve the mode matching, a ramp signal was generated by the first function signal generator (SG1) (Agilent, USA, Model 33500B) to scan the laser wavelength. The transmitted signal was detected by the PD1 and recorded by an oscilloscope (Tektronix, USA, DPO 2024). The second function signal generator (SG2) (Tektronix, USA, AFG 3102) generated two sine-wave signal with the same frequency (15MHz) and an adjustable phase difference. One of them was applied to a fiber-coupled electro-optic modulator ($f$-EOM) (Keyang photonics, China, KY-PM-1550-10-PP-FA) to modulate the laser wavelength. Another was directed to the mixer (Mini-Circuits, USA, ZLW-1) and mixed with the reflected signal that was reflected by the PBS to the PD2. A low-pass filter with an upper cutoff frequency of 1 MHz (Stanford Research Systems, USA, Model SR560) was used after the output of the mixer to acquire a low-frequency signal as an error signal. The error signal was directed to a proportional–integral–derivative (PID) controller (Stanford Research Systems, USA, Model SIM960), which can provide a control signal to adjust the laser wavelength locking it to the cavity mode.

To sweep over a full cavity free-spectral range (FSR) of ~ 0.9 GHz, a 10 Hz ramp signal was applied to the WGM-diode laser, as shown in Fig. 6(a). The transmitted signal and the error signal were recorded using the oscilloscope as shown in Fig. 6(b) and (c), respectively. Based on the FSR and the linewidth of the cavity ($\Delta v$) from Fig. 6(b), the finesse $F$ ($FSR/\Delta v$) is 550. Therefore, a power enhancement factor $G$ of 175 is obtained according to Eq. (1), which means that the detection sensitivity of PAS can be improved by 175 times.

To verify the improved performance of the CEPA sensor system based on the WGM-diode laser, a $C_2H_2$ absorption line located at 1530.98 nm with an intensity of $4.00 \times 10^{-21}$ cm/molecule was selected as target line. The wavelength of the WGM-diode laser was tuned to the target line by means of a wavelength meter (HighFinesse, Germany, WS-6). The laser wavelength was locked to the cavity mode. In order to generate a photoacoustic signal, a fiber-coupled amplitude modulator ($f$-AM) (Photline Technologies, France, MX-LN-10) with a DC bias generated by a high voltage amplifier (Piezomechanik GmbH, Germany, SVR 200-3) and a square-wave of 50 % duty cycle generated by the SG3 (Tektronix, USA, AFG 3102), was employed to modulate the laser intensity before the laser beam entered the collimator. The $f$-AM can provide a DC extinction ratio of 20 dB, which can meet the requirement of the intensity modulation. The frequency of the square-wave was 1781.0 Hz, corresponding to the resonance frequency of the differential photoacoustic cell. The loss of the $f$-EOM and the $f$-AM results in a final incident power of 0.7 mW in front of the F-P cavity. The photoacoustic signal from the differential preamplifier was fed into a lock-in amplifier (LIA) (Stanford Research Systems, USA, Model SR830), which demodulated the signal in the 1-$f$ mode. The reference signal for the LIA was from the TTL signal output of SG3. The parameters of a 12 dB/oct filter slope and 1-s time constant were set for the LIA, corresponding to a detection bandwidth of $\Delta f$ = 0.25 Hz. The demodulated signal from the LIA was recorded by a personal computer and the data was processed with a LabView software program. A verified 500 ppmV $C_2H_2$ gas cylinder was used. The different concentrations of $C_2H_2/N_2$ gas mixtures were produced by a gas dilution system (Environics Inc., USA, Model EN4040).



# 4 Results and discussion

The CEPAM was first filled with 500 ppmV $C_2H_2$. The measurements were carried out at atmospheric pressure and room temperature. The signal amplitudes from the CEPA sensor system are shown in Fig. 7. As a comparison, the signal amplitudes from a CPA sensor without the F-P cavity are also shown in Fig. 7. The CEPA sensor effectively enhanced the

signal amplitude from 44.3 µV to 7,366.8 µV, corresponding to a signal gain factor of 166. Considering the ratio of the powers of incident and transmitted lights is ~95 % for the F-P cavity, the power enhancement factor verified by PAS is 175, which is in excellent agreement with the anticipated value.

In order to evaluate the performance of the CEPA sensor system in terms of minimum detection limit and linearity, pure $N_2$ and five different concentration levels of the $C_2H_2/N_2$ gas mixtures varying from 100 ppmV to 500 ppmV were fed into the

CEPAM. The sensor system was operated at atmospheric pressure and at room temperature. Sixty data points of the CEPA signal were recorded continuously with a 1-s averaging time at each concentration level as shown in Fig. 8(a). With pure $N_2$, the 1σ noise level was found to be 6.61 µV. The scatter of consecutive measurements at a certain concentration level did not depend on the concentration and was in agreement with pure $N_2$. For a 500 ppmV $C_2H_2/N_2$ gas mixture, a signal amplitude of 7,366.8 µV was observed and hence a signal-to-noise ratio (SNR) of 1,114 can be achieved which corresponds to a minimum

detection limit (1σ) of 0.45 ppmV. The plot in Fig. 8(b) is a representation of the same measurements after 60 sensor readings of each concentration step are averaged. This plot confirms the linearity of the sensor response to concentration with a $R$-Square value of >0.9993.

# 5 Conclusions

A CEPA sensor system based on a WGM-diode laser was demonstrated. The WGM-diode laser has an ultra-narrow

linewidth and was used as the excitation source. A cavity-enhanced photoacoustic module was designed to enhance the laser power density inside the optical cavity and the photoacoustic cell, resulting in a signal gain factor of 166, when the laser wavelength was locked on the fundamental optical mode of the F-P cavity by a PDH-locking technique. The combination of the cavity-enhanced absorption spectroscopy and photoacoustic spectroscopy can improve the excitation light power effectively, leading to a significant gain of the photoacoustic signal amplitude. The use of the fiber coupled elements and the

WGM-diode laser have the potential to develop a compact CEPA sensor system with a higher detection sensitivity with respect to the CPA sensor system. A further improvement of the CEPA sensor system can be achieved by using a higher finesse cavity or a higher power laser source.

# 6 Acknowledgements

Lei Dong acknowledges support by National Key R&D Program of China (2017YFA0304203), National Natural Science

Foundation of China (NSFC) (61622503, 61575113, 61805132, 11434007), Changjiang Scholars and Innovative Research




Team in University of Ministry of Education of China (IRT_17R70), 111 project (D18001), Outstanding Innovative Teams of Higher Learning Institutions of Shanxi, Foundation for Selected Young Scientists Studying Abroad, Sanjin Scholar (2017QNSJXZ-04) and Shanxi "1331KSC". Frank K. Tittel acknowledges support by the US National Science Foundation (NSF) ERC MIRTHE award and the Robert Welch Foundation (Grant #C0568).

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





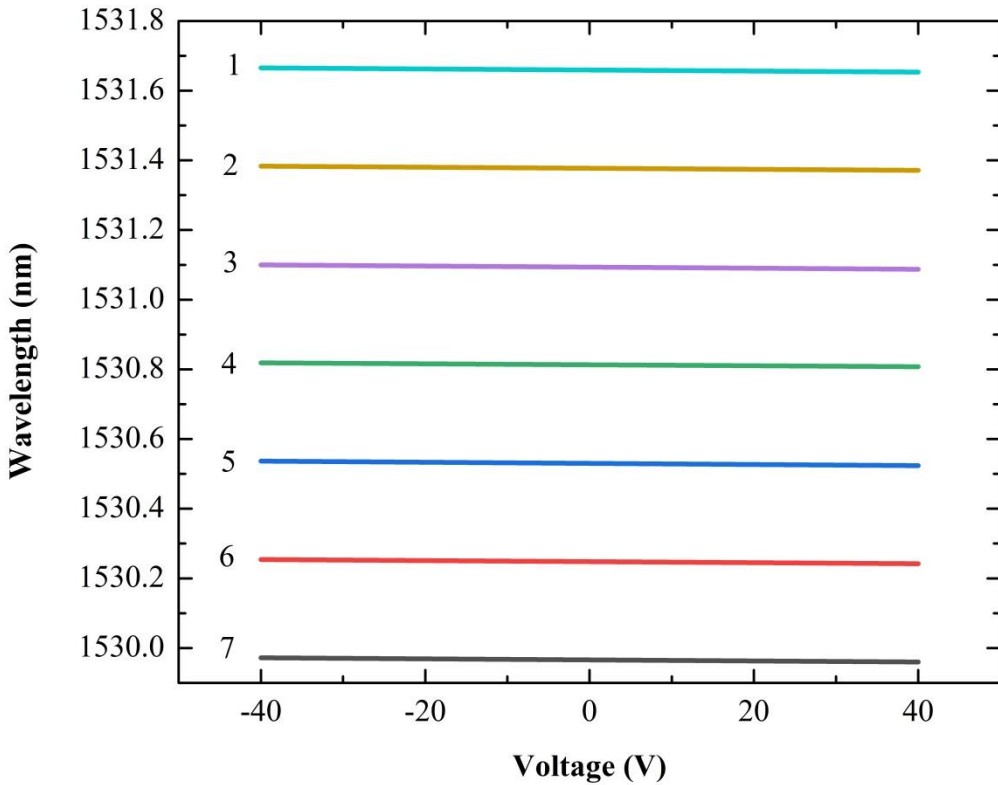

**Figure 1: Tunable wavelength range of the WGM-diode laser for seven different wavelength bands when the laser temperature was fixed and the internal piezoelectric crystal was changed by an external voltage.**



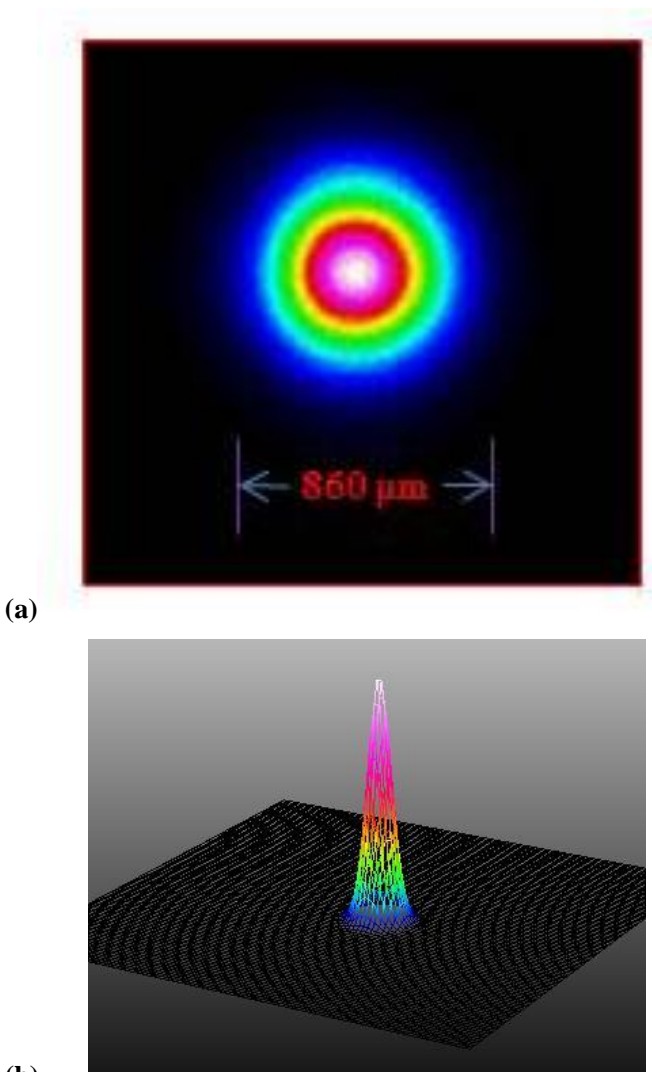

**(a)**

**(b)**

**Figure 2: (a) The two-dimensional intensity distribution of the laser spot; (b) Three-dimensional laser beam profile.**





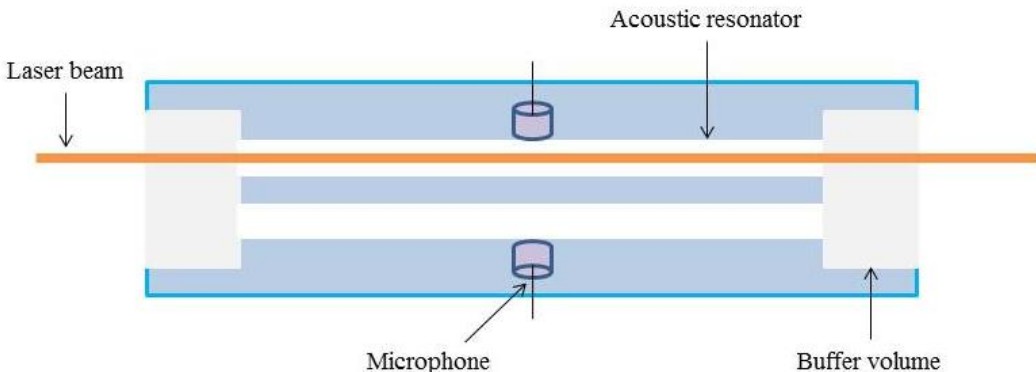

5 **(a)**

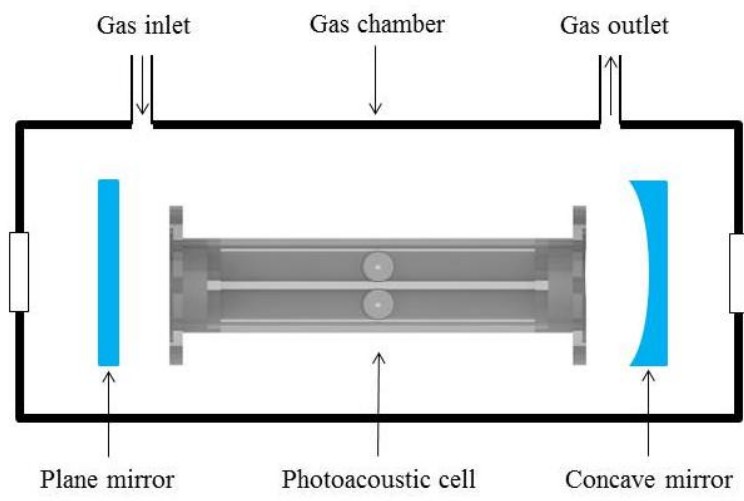

**(b)**

**Figure 3: (a) Schematic diagram of the differential photoacoustic cell; (b) Schematic diagram of the cavity-enhanced photoacoustic module (CEPAM).**




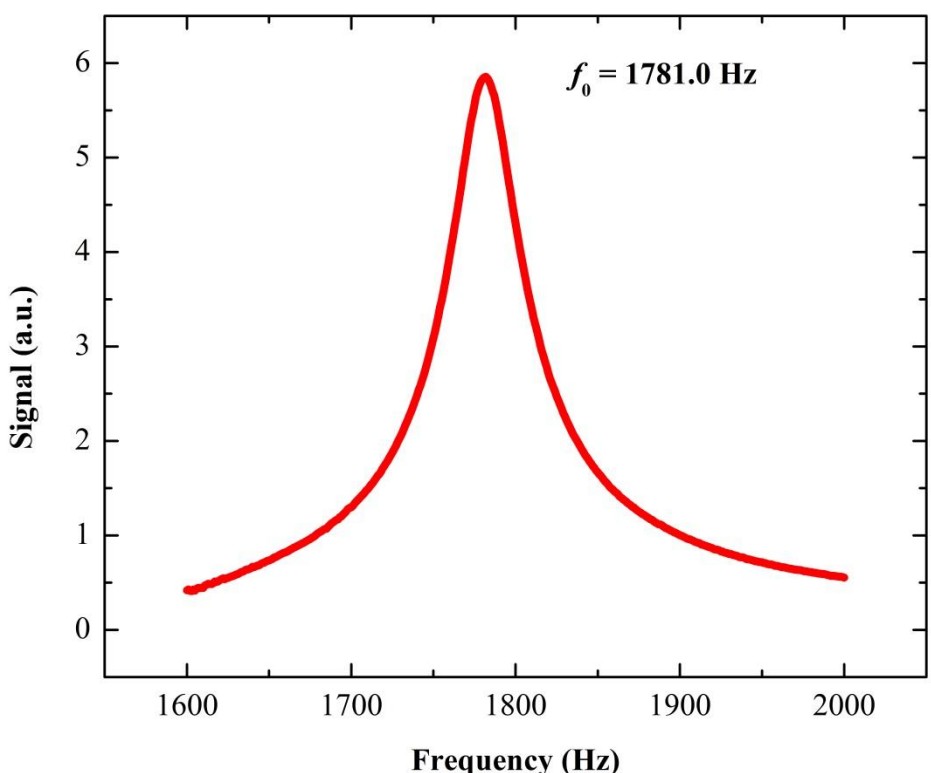

**Figure 4: Frequency response of the differential photoacoustic cell.**



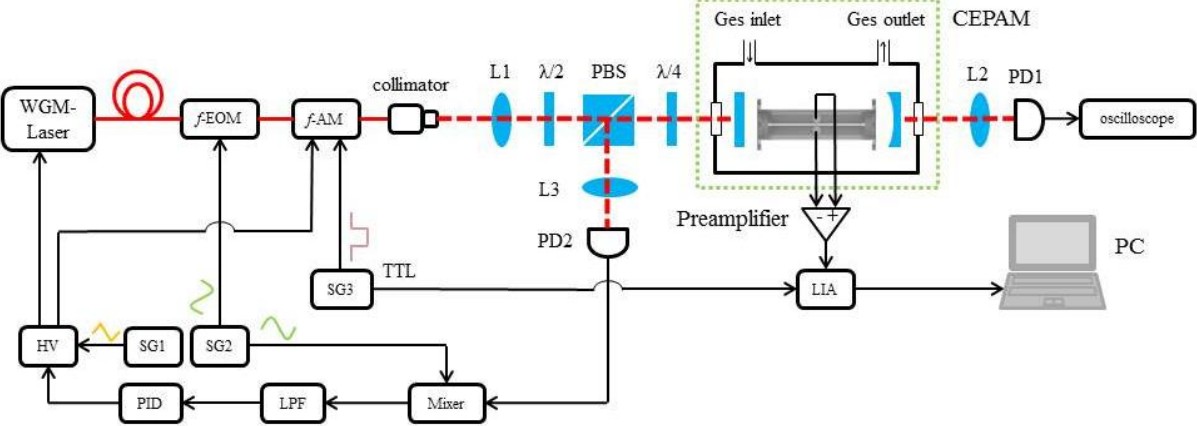

**Figure 5: Experimental setup of the CEPA sensor system: ƒ-EOM, fiber-coupled electro-optic modulator; ƒ-AM, fiber-coupled amplitude modulator; L1, mode-matching lens; λ/2, half wave plate; PBS, polarization beam splitter; λ/4, quarter wave plate; L2 and L3, focusing lenses; PD, photodiode detector; SG, function signal generator; LPF, low-pass filter; PID, proportional–integral– derivative controller; HV, high voltage amplifier; LIA, lock-in amplifier; PC, personal computer.**





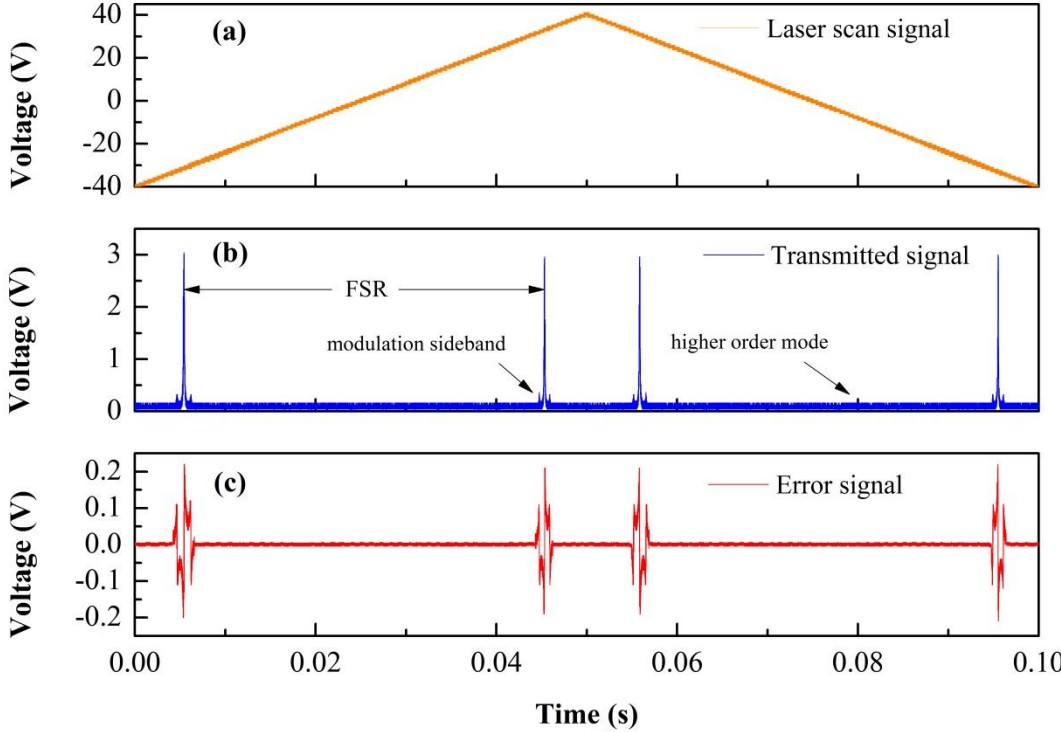

**Figure 6: (a) Laser scan signal; (b) Transmitted signal from the cavity; (c) Error signal from the low-pass filter following the mixer.**

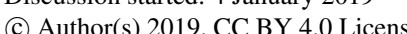



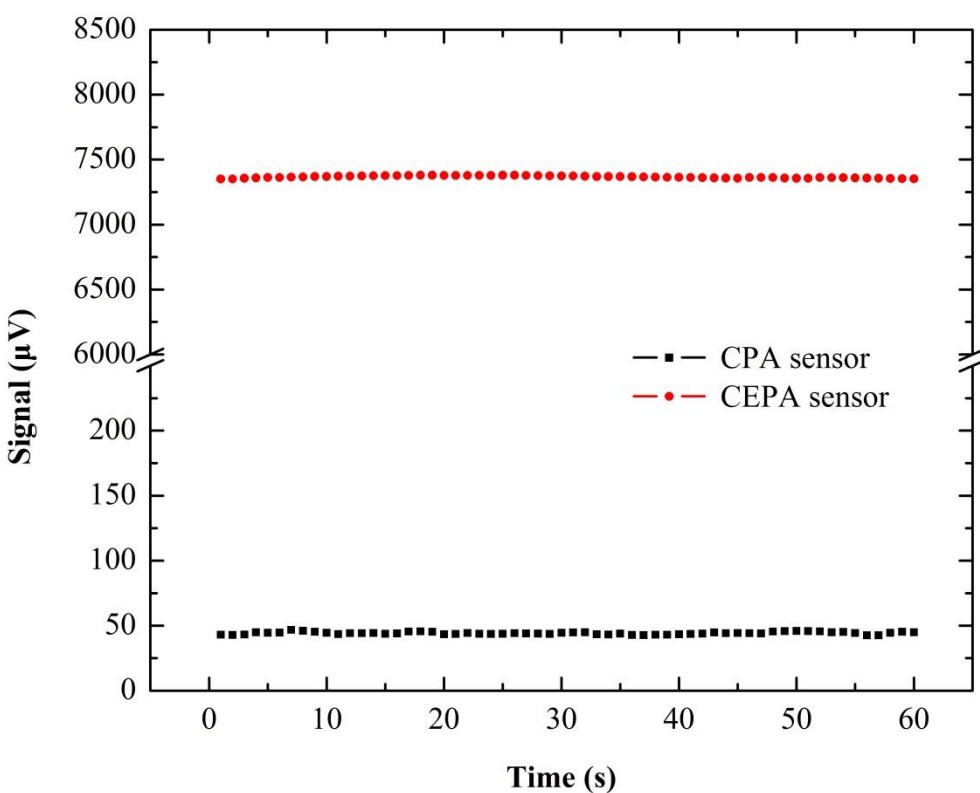

5  **Figure 7: Comparison between the CEPA and CPA signal amplitudes from a 500 ppmV C$_2$H$_2$/N$_2$ gas mixture.**



**(a)**

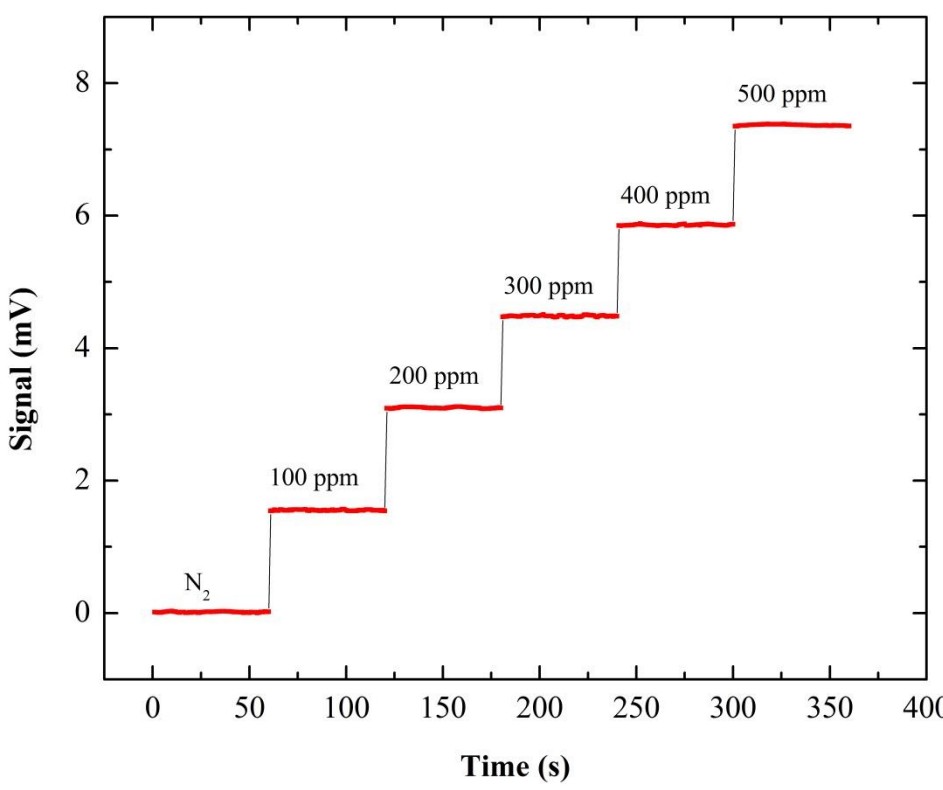





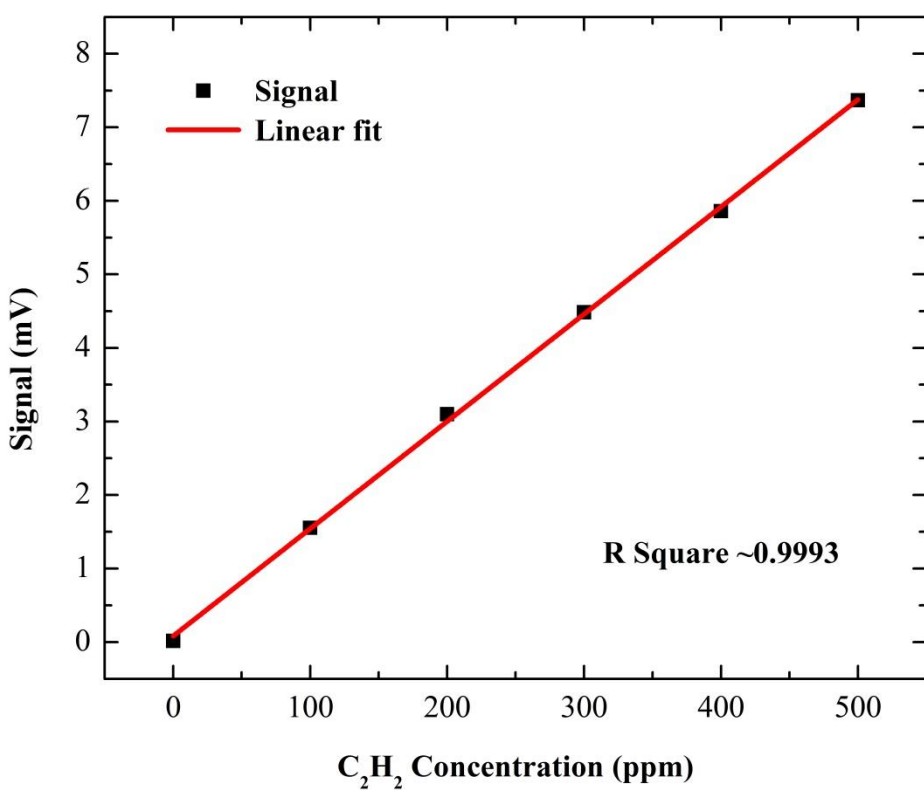

**(b)**

**Figure 8:** (a) CEPA signal at the different C$_2$H$_2$ concentration levels; (b) Linearity of the CEPA sensor system.