# Peer review of "Cavity-enhanced photoacoustic sensor based on a whisperinggallery-mode diode laser"

_Atmospheric Measurement Techniques, 2018_

## Referee Comment (RC1) · Anonymous Referee #1 · 17 Jan 2019

In this manuscript, the authors reported on a very interesting experiment.They implemented an ultra narrow diode laser as exciting source and photoacoustic module coupled with optical build cavity, thereby realizing a cavity-enhanced photoacoustic sensor. To validate the sensor system they selected acetylene as gas target. The authors demonstrate that the cavity-enhanced photoacoustic module was able to increase the photoacoustic signal by a factor of 166 that is comparable with the the power enhancement factor of 175. This result is very interesting for the gas sensing community and the manuscript deserves publication once a few small revisons will be made.

List of revisions:

[Figure]

1) How to switch the diode laser to a different band? By changing its temperature. In this case please show the temperatures related to the different curves shown in Fig.1 2) On page 3 line it should be better described the differential nature of the PAS cell. I suggest the following text: "The gas flow noise and external acoustic disturbances can be effectively suppressed by using a custom transimpedance differential preamplifier. The signal coming from the microphone located in the acoustic resonator not illuminated by the laser beam is subtracted form the one related to the microphone located in the excited resonator and the resulting signal is subsequently amplified. 3) The text on page 3 lines 19-22 describing Fig. 4 should be located after the text describing figure 3b. 4) On page 4 line 28 remove the words: "The parameters of" 5) On page 4 line 31 use certified instead of verified 6) the noise level reported on page 5 line 12 contains too much digits, it should be 6.6 $\mu$V. The same for the signal value reported on page 5 line 14. Better report 7.37 $\mu$V and claim a SNR of 1,110. 7) In the reference list the reference: Patimisco et al, published on Analyst the correct year is 2015 (not 2014).

---

## Referee Comment (RC2) · Anonymous Referee #2 · 18 Jan 2019

This manuscript describes the photoacoustic (PA) detection of acetylene (C2H2) using a compact cavity-enhanced PA sensor in combination with a near-IR whispering-gallery-mode (WGM) diode laser. The paper is clearly written and the results are well documented. However, the paper lacks some major points which should be addressed in a revised version. The following issues need to be taken into account:

General comments 1. What is the exact role of the WGM diode laser, i.e. why WGM ? I understand that the linewidths are of WGM diode lasers are much narrower than those of usual diode lasers but how do measurements at atmospheric pressure benefit from these narrow laser linewidths, i.e. what would be different if a "simple" diode laser

was used ? 2. The detection limit for C2H2 in this paper is not outstanding. In fact, some previous results of PA detection of C2H2 were superior (see e.g.: 33.2 ppb by Yufei Ma et al. in Appl. Phys. Lett 110, 031107 (2017)). A comparison and discussion of previously achieved results for C2H2 using similar techniques is mandatory, e.g. Jingsong Li et al. in Opt. & Laser Techn. 39, 1144 (2007) or Y. Cao et al. in Appl. Phys. B 109, 359 (2012) and others. The pros and cons of the present setup with respect to previously used setups should be clearly stated and discussed in detail. 3. Figures 1, 2, 4 and 7 could eventually be skipped without much loss of information which is given in the text. 4. References: most are references by the authors or co-authors themselves. Additional refs. (incl. those mentioned in point 2) but also others. For example, it is not common to cite just references from own work in a general introductory sentence like on p.1, line 19 (by the way, there it should also be differentiated between Yin et al. 2017a and b).

Minor points 5. The first time WGM-diode laser appears it should be written as Whispering-gallery-mode (WGM)-diode laser (p. 2, line 16). 6. p. 3, line 29: A mode-matching lens (L1),. . . 7. Fig. 7: The color coding in the inset (CPA sensor (black) and CEPA sensor (red)) should be changed to match the order of the measurements, i.e. CEPA on top, CPA at the bottom. However, this figure (see point 3 above) could eventually be skipped.

---

## Author Comment (AC1) · 18 Jan 2019

It's my pleasure to answer your questions.

(1) How to switch the diode laser to a different band? By changing its temperature. In this case please show the temperatures related to the different curves shown in Fig. 1.

It is true that the whispering-gallery-mode diode laser is switched to a different wavelength band by changing its temperature. However, the diode laser is controlled by a software of its own, and the different wavelength bands can be directly selected from the software interface. Therefore, we cannot obtain the corresponding temperatures.

[Figure]

Since questions (2) to (7) are about the English expression and grammar mistakes, we will fix them in the final revised version.

---

## Author Comment (AC2) · 21 Jan 2019

It's my pleasure to answer your questions.

1. What is the exact role of the WGM diode laser, i.e. why WGM? I understand that the linewidths are of WGM diode lasers are much narrower than those of usual diode lasers but how do measurements at atmospheric pressure benefit from these narrow laser linewidths, i.e. what would be different if a "simple" diode laser was used?

A WGM-diode laser has a narrow linewidth, typical <200 Hz. It is true that the linewidth of gas absorption line at atmospheric pressure is typical several GHz. From this

perspective, the measurement can not benefit from these narrow laser linewidths. However, for cavity-enhanced photoacoustic spectroscopy, laser wavelength must be locked to the resonant mode of the optical cavity to allow the energy to build up inside the cavity. The higher the finesse of the optical cavity is, the narrower the linewidth of the cavity mode will be. The WGM-diode laser with a narrow linewidth is very helpful to realize the wavelength locking, especially in the condition of a high finesse optical cavity. In our next work, we plan to design a high finesse cavity (e.g.: 5000 or 10000). Therefore, we must use an ultra-narrow linewidth laser. If a "simple" diode laser was used, it will be a challenge to complete the locking between the resonant mode of the optical cavity and the laser wavelength.

2. The detection limit for $C_2H_2$ in this paper is not outstanding. In fact, some previous results of PA detection of $C_2H_2$ were superior (see e.g.: 33.2 ppb by Yufei Ma et al. in Appl. Phys. Lett 110, 031107 (2017)). A comparison and discussion of previously achieved results for $C_2H_2$ using similar techniques is mandatory, e.g. Jingsong Li et al. in Opt. & Laser Techn. 39, 1144 (2007) or Y. Cao et al. in Appl. Phys. B 109, 359 (2012) and others. The pros and cons of the present setup with respect to previously used setups should be clearly stated and discussed in detail.

There are three reasons why our detection limit is inferior compared to that from Yufei Ma et al.. (1). In our experiment, the absorption line of acetylene that we selected was 6531.76 cm-1, which was different from the above-mentioned paper (Yufei Ma et al. in Appl. Phys. Lett 110, 031107 (2017) (6534.37 cm-1). The line strength of 6531.76 cm-1 is $\sim$3 times lower than that of 6534.37 cm-1. (2) We did not move the cavity mode to the top of the absorption line. In this way, we were detecting the $C_2H_2$ using its spectral wing, which resulted in a sensitivity loss of $\sim$3 times. (3) With the optical cavity, our effective optical power was 116 mW. But Yufei Ma used an EDFA to boost the laser power to 1500 mW.

In fact, we are more concerned with the signal enhancement from the cavity in this manuscript. Therefore, we did not pay more attention to obtaining a better minimum

detection limit. After selecting the same absorption line and moving the cavity mode to the top of the absorption line, we expect to achieve a comparative detection limit with that from Yufei Ma.

As for the detection limits achieved in the above-mentioned papers (Jingsong Li et al. in Opt. & Laser Techn. 39, 1144 (2007) and Y. Cao et al. in Appl. Phys. B 109, 359 (2012)), they were 10 ppmV and 2 ppmV, respectively, which are inferior to the detection limit that we reported.

Furthermore, the cavity-enhanced photoacoustic technique we used is different from those in the above-mentioned papers, which used conventional photoacoustic spectroscopy or quartz-enhanced photoacoustic spectroscopy (QEPAS) to detect C2H2. We admit that the use of an optical cavity makes the sensor system more complicated, but it has a potential to further improve the detect limit if a higher finesse cavity is employed since the detect sensitivity is proportional to the excitation optical power. In our next work, we will use a high finesse cavity to improve the signal-to-noise ratio (SNR), thus making the detection limit better.

We will make a detail discussion regarding the pros and cons of the present setup with respect to previously used setups in our revised version.

3. Figures 1, 2, 4 and 7 could eventually be skipped without much loss of information which is given in the text.

Figures 1 and 2 describe the excellent characteristics of the WGM-diode laser about wavelength tunable ability and intensity distribution. We would like to leave Fig. 1 and 2 since they are two necessary indexes for the selection of an excitation source in PAS and CEAS. We will delete the Figure 4. Figure 7 shows a comparison of the photoacoustic signal between the CPA sensor and the CEPA sensor. The signal enhancement can be observed clearly from Fig. 7. And the stability of a constant signal can also be shown. We suggested to leave this figure.

Since questions 4-7 are about the references and minor points, we will fix them in the final revised version.

---

## Author Response (AR1)

**Responses to reviewers' and editor's comments**

**Dear Editor:**

We thank you and the reviewers for your comments and contributions to improve this manuscript. Please find enclosed our answers to the reviewers' and editor's comments. We have addressed all issues, and revised our manuscript, accordingly.

Please note: reviewers' comments are in black; our comments are in *red italic*; the original paper text in red and revised text in blue.

**Reviewer Comments:**

**Reviewer 1:**

In this manuscript, the authors reported on a very interesting experiment. They implemented an ultra narrow diode laser as exciting source and photoacoustic module coupled with optical build cavity, thereby realizing a cavity-enhanced photoacoustic sensor. To validate the sensor system they selected acetylene as gas target. The authors demonstrate that the cavity-enhanced photoacoustic module was able to increase the photoacoustic signal by a factor of 166 that is comparable with the power enhancement factor of 175. This result is very interesting for the gas sensing community and the manuscript deserves publication once a few small revisions will be made.

**List of revisions:**

1) How to switch the diode laser to a different band? By changing its temperature. In this case please show the temperatures related to the different curves shown in Fig.1.

It is true that the whispering-gallery-mode diode laser is switched to a different wave- length band by changing its temperature. However, the diode laser is controlled by a software of its own, and the different wavelength bands can be directly selected from the software interface. Therefore, we cannot obtain the corresponding temperatures.

2) On page 3 line it should be better described the differential nature of the PAS cell. I suggest the following text: "The gas flow noise and external acoustic disturbances can be effectively suppressed by using a custom transimpedance differential preamplifier. The signal coming from the microphone located in the acoustic resonator not illuminated by the laser beam is subtracted from the one related to the microphone located in the excited resonator and the resulting signal is subsequently amplified.

**Many thanks for the reviewer's suggestion. We replaced the following sentence:**

"The gas flow noise and external acoustic disturbances can be effectively suppressed by using a custom transimpedance differential preamplifier to amplify the signal from the two microphones, due to the fact that only one of the two acoustic resonators is excited by the laser,"

**with:**

"The gas flow noise and external acoustic disturbances can be effectively suppressed by using a custom transimpedance differential preamplifier. The signal coming from the microphone located in the acoustic resonator not illuminated by the laser beam is subtracted from the one related to the microphone located in the excited resonator and the resulting signal is subsequently amplified."

3) The text on page 3 lines 19-22 describing Fig. 4 should be located after the text describing figure 3b.

We deleted the Fig. 4 according to Referee #2's suggestion.

4) On page 4 line 28 remove the words: "The parameters of".

Many thanks for the reviewer's suggestion. We replaced the following sentence:

"The parameters of a 12 dB/oct filter slope and 1-s time constant were set for the LIA, corresponding to a detection bandwidth of  $\Delta f = 0.25$  Hz."

with:

"A 12 dB/oct filter slope and a 1-s time constant were set for the LIA, corresponding to a detection bandwidth of  $\Delta f = 0.25$  Hz."

5) On page 4 line 31 use certified instead of verified.

Many thanks for the reviewer's suggestion. We replaced the following sentence:

"A verified 500 ppmV C2H2 gas cylinder was used."

with:

"A certified 500 ppmV C2H2 gas cylinder was used."

6) the noise level reported on page 5 line 12 contains too much digits, it should be 6.6  $\mu$ V. The same for the signal value reported on page 5 line 14. Better report 7.37  $\mu$ V and claim a SNR of 1,110.

We changed the values according to the suggestion.

7) In the reference list the reference: Patimisco et al, published on Analyst the correct year is 2015 (not 2014).

Many thanks for the reviewer's suggestion. We replaced the following sentence:

"Patimisco, P., Borri, S., Galli, I., Mazzotti, D., Giusfredi, G., Akikusa, N., Yamanishi, M., Scamarcio, G., Natale, P. D., and Spagnolo, V.: High finesse optical cavity coupled with a quartz-enhanced photoacoustic spectroscopic sensor, Analyst, 140, 736-743, 2014."

with:

"Patimisco, P., Borri, S., Galli, I., Mazzotti, D., Giusfredi, G., Akikusa, N., Yamanishi, M., Scamarcio, G., Natale, P. D., and Spagnolo, V.: High finesse optical cavity coupled with a

quartz-enhanced photoacoustic spectroscopic sensor, Analyst, 140, 736-743, 2015."

**Reviewer 2:**

This manuscript describes the photoacoustic (PA) detection of acetylene (C2H2) us- ing a compact cavity-enhanced PA sensor in combination with a near-IR whispering- gallery-mode (WGM) diode laser. The paper is clearly written and the results are well documented. However, the paper lacks some major points which should be addressed in a revised version. The following issues need to be taken into account:

**General comments**

1. What is the exact role of the WGM diode laser, i.e. why WGM ? I understand that the linewidths are of WGM diode lasers are much narrower than those of usual diode lasers but how do measurements at atmospheric pressure benefit from these narrow laser linewidths, i.e. what would be different if a "simple" diode laser was used?

A WGM-diode laser has a narrow linewidth, typical <200 Hz. It is true that the linewidth of gas absorption line at atmospheric pressure is typical several GHz. From this perspective, the measurement cannot benefit from these narrow laser linewidths. However, for cavity-enhanced photoacoustic spectroscopy, laser wavelength must be locked to the resonant mode of the optical cavity to allow the energy to build up inside the cavity. The higher the finesse of the optical cavity is, the narrower the linewidth of the cavity mode will be. The WGM-diode laser with a narrow linewidth is very helpful to realize the wavelength locking, especially in the condition of a high finesse optical cavity. In our next work, we plan to design a high finesse cavity (e.g.: 5000 or 10000). Therefore, we must use an ultra-narrow linewidth laser. If a "simple" diode laser was used, it will be a challenge to complete the locking between the resonant mode of the optical cavity and the laser wavelength.

2. The detection limit for C2H2 in this paper is not outstanding. In fact, some previous results of PA detection of C2H2 were superior (see e.g.: 33.2 ppb by Yufei Ma et al. in Appl. Phys. Lett 110, 031107 (2017)). A comparison and discussion of previously achieved results for C2H2 using similar techniques is mandatory, e.g. Jingsong Li et al. in Opt. & Laser Techn. 39, 1144 (2007) or Y.Cao et al. in Appl. Phys. B 109, 359 (2012) and others. The pros and cons of the present setup with respect to previously used setups should be clearly stated and discussed in detail.

There are three reasons why our detection limit is inferior compared to that from Yufei Ma et al. (1). In our experiment, the absorption line of acetylene that we selected was  $6531.76 \text{ cm}^{-1}$ , which was different from the above-mentioned paper (Yufei Ma et al. in Appl. Phys. Lett 110, 031107 (2017) ( $6534.37 \text{ cm}^{-1}$ ). The line strength of  $6531.76 \text{ cm}^{-1}$  is  $\sim 3$  times lower than that of  $6534.37 \text{ cm}^{-1}$ . (2) We did not move the cavity mode to the top of the absorption line. In this way, we were detecting the  $C_2H_2$  using its spectral wing, which resulted in a sensitivity loss of  $\sim 3$  times. (3) With the optical cavity, our effective optical power was 116 mW. But Yufei Ma used an EDFA to boost the laser power to 1500 mW.

In fact, we are more concerned with the signal enhancement from the cavity in this manuscript. Therefore, we did not pay more attention to obtaining a better minimum detection limit. After selecting the same absorption line and moving the cavity mode to the top of the absorption line, we expect to achieve a comparative detection limit with that from Yufei Ma.

As for the detection limits achieved in the above-mentioned papers (Jingsong Li et al. in Opt. & Laser Techn. 39, 1144 (2007) and Y. Cao et al. in Appl. Phys. B 109, 359 (2012)), they were 10 ppmV and 2 ppmV, respectively, which are inferior to the detection limit that we reported.

Furthermore, the cavity-enhanced photoacoustic technique we used is different from those in the above-mentioned papers, which used conventional photoacoustic spectroscopy or quartz-enhanced photoacoustic spectroscopy (QEPAS) to detect  $C_2H_2$ . We admit that the use of an optical cavity makes the sensor system more complicated, but it has a potential to further improve the detect limit if a higher finesse cavity is employed since the detection sensitivity is proportional to the excitation optical power. In our next work, we will use a high finesse cavity to improve the signal-to-noise ratio (SNR), thus making the detection limit better.

**In order to clarify it, we added the sentences in Results and discussion part:**

"In 2017, Yufei Ma et al. used QEPAS technique to detect  $C_2H_2$  and obtain a detection sensitivity of 33.2 ppb (Ma et al., 2017), which is 13 times better than this CEPA sensor. The possible reasons are (1) The CEPA sensor employed the acetylene absorption line located at  $6531.76 \text{ cm}^{-1}$ , which was different from the Yufei Ma's line at  $6534.37 \text{ cm}^{-1}$ . The line strength of  $6531.76 \text{ cm}^{-1}$  is ~3 times lower than that of  $6534.37 \text{ cm}^{-1}$ . (2) The cavity mode was not moved to the top of the absorption line. In this way, the  $C_2H_2$  spectral wing was detected, which resulted in a sensitivity loss of ~3 times. (3) The effective optical power in the optical cavity was 116 mW. But Yufei Ma et al. used an EDFA to boost the laser power to 1,500 mW. In fact, the signal enhancement achieved from the cavity is more important in this research. A comparative detection limit can be expected after selecting the same absorption line and setting the cavity mode to the top of the absorption line.

The CEPA technique is very different from the CPA technique. It is true that the use of an optical cavity makes the sensor system more complicated. However, the CEPA technique has the potential to further improve the detect limit if a higher finesse cavity is employed since the detection sensitivity is proportional to the excitation optical power."

**and added the reference:**

"Ma, Y., He, Y., Zhang, L., Yu, X., Zhang, J., Sun, R., and Tittel, F. K.: Ultra-high sensitive acetylene detection using quartz-enhanced photoacoustic spectroscopy with a fiber amplified diode laser and a 30.72 kHz quartz tuning fork, Applied Physics Letters, 110, 031107, 2017."

3. Figures 1, 2, 4 and 7 could eventually be skipped without much loss of information which is given in the text.

Figures 1 and 2 describe the excellent characteristics of the WGM-diode laser about wavelength tunable ability and intensity distribution. We would like to leave Fig. 1 and 2

since they are two necessary indexes for the selection of an excitation source in PAS and CEAS. We will delete the Figure 4. Figure 7 shows a comparison of the photoacoustic signal between the CPA sensor and the CEPA sensor. The signal enhancement can be observed clearly from Fig. 7. And the stability of a constant signal can also be shown. We suggested to leave this figure.

4. References: most are references by the authors or co-authors themselves. Additional refs. (incl. those mentioned in point 2) but also others. For example, it is not common to cite just references from own work in a general introductory sentence like on p.1, line 19 (by the way, there it should also be differentiated between Yin et al. 2017a and b).

Many thanks for the reviewer's suggestion. We replaced the following references:

"Yin, X., Dong, L., Wu, H., Zheng, H., Ma, W., Zhang, L., Yin, W., Xiao, L., Jia, S., and Tittel, F. K.: Sub-ppb nitrogen dioxide detection with a large linear dynamic range by use of a differential photoacoustic cell and a 3.5 W blue multimode diode laser, Sensors and Actuators B: Chemical, 247, 329-335, 2017."

"He, Q., Lou, M., Zheng, C., Ye, W., Wang, Y., and Tittel, F. K.: Repetitively mode-locked cavity-enhanced absorption spectroscopy (rml-ceas) for near-infrared gas sensing, Sensors, 17, 2792, 2017."

**with:**

"Siciliani, d. C. M., Viciani, S., Borri, S., Patimisco, P., Sampaolo, A., Scamarcio, G., Natale, P. D., D'Amato, F., and Spagnolo, V.: Widely-tunable mid-infrared fiber-coupled quartz-enhanced photoacoustic sensor for environmental monitoring, Optics Express, 22, 28222-28231, 2014."

"Yi, H., Wu, T., Wang, G., Zhao, W., Fertein, E., Coeur, C., Gao, X., Zhang, W., and Chen, W.: Sensing atmospheric reactive species using light emitting diode by incoherent broadband cavity enhanced absorption spectroscopy, Optics Express, 24, A781-A790, 2016."

**Minor points**

5. The first time WGM-diode laser appears it should be written as Whispering-gallery-mode (WGM)-diode laser (p. 2, line 16).

**Done.**

6. p. 3, line 29: A mode- matching lens (L1),...

**Done.**

7. Fig. 7: The color coding in the inset (CPA sensor (black) and CEPA sensor (red) should be changed to match the order of the measurements, i.e. CEPA on top, CPA at the bottom. However, this figure (see point 3 above) could eventually be skipped.

**Done.**